

# Modified STOP-Bang questionnaire for detecting obstructive sleep apnea in individuals with a body mass index below 35 kg/m²

Napassorn Sinsopa[*], Viriya Tripakornkusol[*], Sittichai Khamsai and Kittisak Sawanyawisuth

Department of Medicine, Khon Kaen University, Khon Kaen, Thailand
[*] These authors contributed equally to this work.

## ABSTRACT

**Background.** Obstructive sleep apnea (OSA) is a common clinical condition. Due to its high prevalence, the waiting list for polysomnography is often long. A screening tool is needed to identify individuals at high risk for OSA who should undergo polysomnography. The STOP-Bang questionnaire is a widely used screening tool; however, it may require modification for individuals with a body mass index (BMI) below 35 kg/m². This study aimed to evaluate whether the STOP-Bang questionnaire should be modified for patients with a BMI under 35 kg/m².

**Methods.** This retrospective analytical study included adult patients suspected of having OSA who underwent polysomnography. Exclusion criteria included pregnancy and a BMI over 35 kg/m² or meeting criteria for bariatric surgery. Patients were categorized into OSA and non-OSA groups. Logistic regression analysis was used to assess the predictive value of STOP-Bang factors for OSA.

**Results.** A total of 188 patients were included, of whom 158 (84.04%) were diagnosed with OSA. Among the eight STOP-Bang criteria, only age was independently associated with OSA after adjustment for other variables (adjusted odds ratio: 1.04; 95% CI [1.02–1.08]). Optimal cut-off points for predicting OSA were identified as age ≥ 40 years (sensitivity: 84.18%), BMI ≥ 23 kg/m² (sensitivity: 82.91%), and neck circumference ≥ 35 cm (sensitivity: 86.08%). A modified STOP-Bang score incorporating these cut-offs showed improved sensitivity at a score of 3: 93.0% for apnea-hypopnea index (AHI) ≥ 5 events/hr, 95.9% for AHI ≥ 15 events/hr, and 97.6% for AHI ≥ 30 events/hr. In comparison, the original STOP-Bang score of 3 had sensitivities of 50.0%, 53.6%, and 56.1% for AHI ≥ 5 events/hr, AHI ≥ 15 events/hr, and AHI > 30 events/hr, respectively.

**Conclusions.** The STOP-Bang questionnaire may require modification for individuals with a BMI below 35 kg/m² who are suspected of having OSA. Revised cut-off values for age, neck circumference, and BMI—40 years, 35 cm, and 23 kg/m², respectively—may enhance its diagnostic performance.

Corresponding authors
Sittichai Khamsai, sittikh@kku.ac.th
Kittisak Sawanyawisuth, kittisak@kku.ac.th

## INTRODUCTION

Obstructive sleep apnea (OSA), defined by an apnea-hypopnea index (AHI) of five or more per hour on polysomnography, is a common disease in clinical practice. A systematic review showed that the global prevalence of OSA was 54% and that its prevalence was over 50% in all age groups (*De Araujo Dantas et al., 2023*). In addition to its high prevalence, OSA has been shown to be associated with several cardiovascular diseases (*Khamsai et al., 2021c*; *Khamsai et al., 2021a*; *Soontornrungsun et al., 2020*; *Khamsai et al., 2021b*). The 2021 European Society of Cardiology stated that OSA is strongly associated with hypertension, stroke, heart failure, and atrial fibrillation (*Visseren et al., 2021*). Treatment with a continuous positive airway pressure (CPAP) machine is effective and beneficial.

The high incidence of OSA has resulted in a long waiting list for polysomnography. Undiagnosed or untreated OSA may increase the risk of future cardiovascular diseases or other morbidities. Adult patients with OSA are at risk for cardiovascular events, with a relative risk of 1.96 (95% CI [1.90–2.02]) compared with controls (*Albertsen et al., 2024*). Additionally, the adjusted odds ratio for hospitalization was 1.82 times (95% CI [1.77–1.87]) higher in undiagnosed OSA compared with non-OSA (*Kirk et al., 2023*). Polysomnography, which comprises an electroencephalogram, an electrooculogram, an electromyogram, an electrocardiogram, pulse oximetry, airflow, and respiratory effort, is a diagnostic test for OSA that identifies apnea and hypopnea during sleep (*Rundo & Downey, 2019*). As polysomnography is time-consuming and expensive, using a screening tool to select high-risk individuals for polysomnography is needed. A previous study showed that the STOP-Bang questionnaire has a good sensitivity of 83.6% for detecting OSA (*Chung et al., 2012*). The STOP-Bang questionnaire stands for snoring, tiredness during the day, observed apnea, high blood pressure, a body mass index (BMI) of 35 kg/m$^2$, an age of over 50 years, a neck circumference of over 40 cm, and male sex. A cut point of three indicates OSA, with an AHI $\geq$ 5 events/hour. However, in that study, 34 out of 177 patients (19.21%) had a body mass index of more than 35 kg/m$^2$. Patients with body mass index of less than 35 kg/m$^2$ may have different clinical features. A modified STOP-Bang questionnaire for individuals with suspected OSA with body mass index of less than 35 kg/m$^2$ may be needed and this modified version may provide a more sensitive screening tool. This study aimed to evaluate whether the STOP-Bang questionnaire needed to be modified for patients with a body mass index of less than 35 kg/m$^2$.

## MATERIALS & METHODS

This was a retrospective, analytical study conducted at Srinagarind Hospital, Khon Kaen University, Thailand. The inclusion criteria were adult patients who were suspected of having OSA tested by polysomnography. Those who were pregnant; met the criteria for bariatric surgery with a body mass index of more than 35 kg/m$^2$; had other sleep disorders, such as restless leg syndrome or narcolepsy; had an incomplete questionnaire or missing data of over 50%; or had a failed polysomnography were excluded. Suspected OSA was described by a previous study as including a history of snoring, stopped breathing while sleeping, fatigue, sleepiness, or being diagnosed with obesity, hypertension,

or cardiovascular diseases (*Jonassen et al., 2022*). These patients were requested for polysomnography by their attending physicians. This study was a part of an OSA project at Khon Kaen University, Thailand. The study protocol was approved by the Khon Kaen University Ethics Committee in Human Research, Thailand (HE641504). The study period was between March and December 2023.

Patients who were suspected of having OSA at the outpatient department were enrolled in the study. Those who had an AHI ≥ 5 events/hour were diagnosed with OSA, while those with an AHI ≤ 5 events/hour were defined as non-OSA (*Jonassen et al., 2022*). STOP-Bang criteria were applied to both the OSA and non-OSA groups, including a history of snoring (S), a history of being tired or having fatigue during the day (T), a history of observed apnea (O), a history of hypertension (P), body mass index (B), age (A), neck circumference (N), and male gender (G). The neck circumference was measured in cm at the cricothyroid membrane. A summation of "yes" answers to each acronym of the STOP-Bang questions was the STOP-Bang score (*Miller et al., 2018*). Note that the STOP-Bang questionnaire had a sensitivity of 94.44% and a specificity of 20.59% for OSA detection with an AHI ≥ 5 events/hour, while the KR20 was 0.407 (*Miller et al., 2018*).

## Sample size calculation

A previous study found that 68.92% of patients had OSA (*Chung et al., 2008*); the estimated proportion of OSA in this study, which had a lower body mass index, was 55%. With a power of 95% and a confidence level of 95%, the estimated study population was 154 patients.

## Statistical analyses

Patients were categorized into two groups: an OSA group and a non-OSA group. The STOP-Bang factors were compared between both groups using inferential statistics. For numerical factors, data were presented as median (range), while numbers (proportions) were presented for categorical factors. Differences in numerical factors between the two groups were tested using the Wilcoxon Rank-Sum test, while the Fisher Exact test was used to compare the differences between the two proportions.

The original STOP-Bang factors were calculated to predict the presence of OSA by using logistic regression analysis with both univariable and multivariable methods, as previously described by the original STOP-Bang study (*Chung et al., 2008*). Each factor of the STOP-Bang criteria was calculated for an unadjusted odds ratio with a 95% confidence interval by a univariable logistic regression analysis. All STOP-Bang factors were then put into the multivariable logistic regression analysis to calculate adjusted odds ratios with 95% confidence intervals. The Hosmer-Lemeshow test was used to evaluate the goodness of fit of the regression model. A Hosmer-Lemeshow chi-square of more than 0.05 indicates a goodness of fit. The numerical STOP-Bang factors, including body mass index, age, and neck circumference, were executed for sensitivity and specificity using the area under a receiver operating characteristic (ROC) curve. An appropriate cut point for these three numerical factors was chosen to create the modified STOP-Bang score for each patient. The chosen cut point was selected to provide a sensitivity of 80% or more.

**Table 1  The STOP-Bang criteria of patients screened for obstructive sleep apnea (OSA) categorized by presence of OSA.**

| Factors | Non OSA n = 30 | OSA n = 158 | p value |
|---|---|---|---|
| Snoring | 22 (73.33) | 136 (86.08) | 0.101 |
| Tired during the day | 16 (53.33) | 73 (46.20) | 0.551 |
| Stop breathing | 4 (13.33) | 40 (25.48) | 0.239 |
| Hypertension | 12 (40.00) | 101 (63.92) | 0.037 |
| Body mass index, kg/m$^2$ | 25.2 (16.2–34.4) | 26.2 (16.5–34.9) | 0.266 |
| Age, year | 51 (20–80) | 58 (19–79) | 0.006 |
| Neck circumference, cm | 36 (30–45) | 38 (28–48) | 0.082 |
| Male sex | 17 (56.67) | 71 (44.94) | 0.318 |
| STOP-Bang score | 3 (1–5) | 4 (0–8) | 0.119 |

**Notes.**
Data presented as number (percentage) or median (range).

Both the original and modified STOP-Bang scores were tested as predictors for OSA by logistic regression analysis. The odds ratio with a 95% confidence interval for the original and modified STOP-Bang scores was computed, as well as the area under the ROC curve, to predict OSA by three different settings: AHI $\geq$ 5, $\geq$ 15, and $\geq$ 30 events/hour (*Chung et al., 2008*; *De Menezes Júnior et al., 2022*; *De Menezes-Júnior et al., 2023*). Diagnostic properties, including sensitivity, specificity, positive predictive value (PPV), negative predictive value (NPV), and Youden score, were executed for each total STOP-Bang score by both original and modified STOP-Bang criteria (*Chung et al., 2008*; *De Menezes Júnior et al., 2022*; *De Menezes-Júnior et al., 2023*). These calculations were repeated for three different OSA severities: AHI $\geq$ 5, $\geq$ 15, and $\geq$ 30 events/hour. The statistical analyses were performed using STATA software version 18.0 (College Station, TX, USA).

## RESULTS

There were 188 patients included in the study. Of those, 158 patients (84.04%) were diagnosed with OSA. Two factors in the STOP-Bang criteria were significantly different between the non-OSA and OSA groups: hypertension and age, as shown in Table 1. The OSA group had a significantly higher proportion of hypertension than the non-OSA group (63.92% *vs.* 40.00%; $p = 0.024$), while the OSA group was older than the non-OSA group (58 *vs.* 51 years; $p = 0.006$).

Among the eight factors in the STOP-Bang criteria, both hypertension and age were significantly associated with the presence of OSA by univariable logistic regression analysis, with unadjusted odds ratios of 2.51 and 1.03, respectively (Table 2). After adjusting for other variables, only age was independently associated with the presence of OSA, with an adjusted odds ratio of 1.04 (95% CI [1.02–1.08]). The Hosmer-Lemeshow chi-square of the model was 4.42 ($p = 0.817$), indicating a goodness of fit for the model. The area under the ROC curve for age to predict the presence of OSA was 65.80% (95% CI [54.74%–76.88%]), as shown in Fig. 1.

**Table 2 Unadjusted and adjusted odds ratio to predict presence of obstructive sleep apnea by using the STOP-Bang criteria.**

| Factors | Unadjusted odds ratio (95% confidence interval) | Adjusted odds ratio (95% confidence interval) |
|---|---|---|
| Snoring | 2.24 (0.89, 5.67) | 2.70 (0.95, 7.70) |
| Tired | 0.75 (0.34, 1.64) | 0.81 (0.34, 1.95) |
| Observed apnea | 2.22 (0.73, 6.75) | 1.96 (0.59, 6.49) |
| Hypertension | 2.51 (1.11, 5.62) | 2.01 (0.84, 4.84) |
| Body mass index | 1.05 (0.96, 1.16) | 1.07 (0.95, 1.23) |
| Age | 1.03 (1.01, 1.06) | 1.04 (1.02, 1.08) |
| Neck circumference | 1.09 (0.98, 1.21) | 1.02 (0.87, 1.21) |
| Male sex | 0.62 (0.28, 1.37) | 0.68 (0.24, 1.93) |

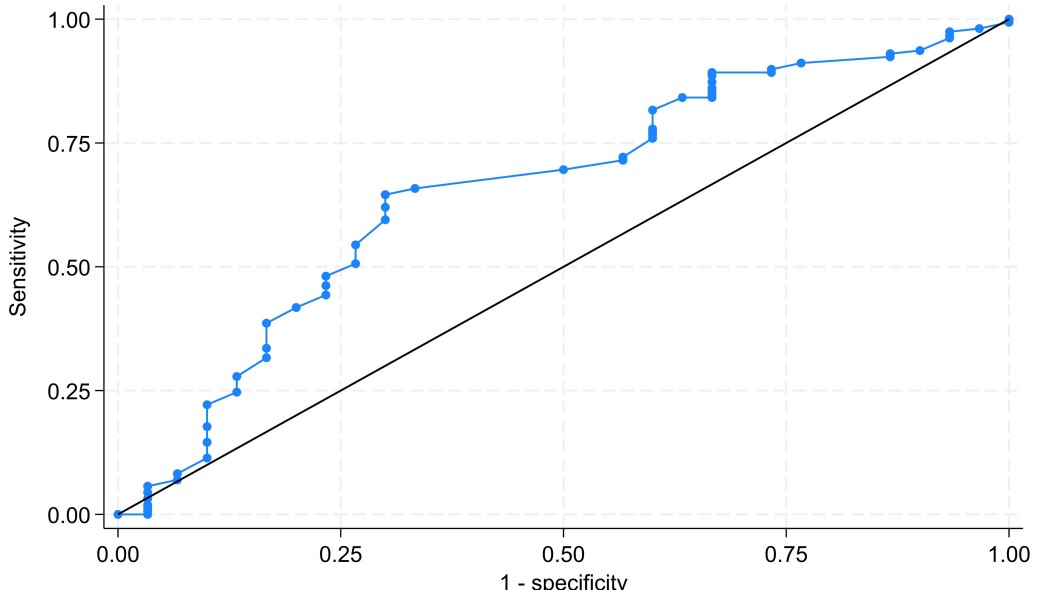

**Figure 1 ROC curve of age to predict presence of obstructive sleep apnea.** The area under the ROC curve was 65.80%. ROC, receiver operating characteristic.

**Table 3 Sensitivity and specificity of predictors in the STOP-Bang criteria of obstructive sleep apnea.**

| Factors | Sensitivity | Specificity |
|---|---|---|
| Age of 40 years | 84.18% | 33.33% |
| Body mass index of 23 kg/m$^2$ | 82.91% | 30.00% |
| Neck circumference of 35 cm | 86.08% | 33.33% |

The cut-off points of three numerical factors used to predict the presence of OSA, including age, body mass index, and neck circumference, are shown in Table 3. An age of 40 years had a sensitivity of 84.18%, while a body mass index of 23 kg/m$^2$ had a sensitivity of 82.91%. A neck circumference of 35 cm had the highest sensitivity at 86.08%.

**Table 4   Odds ratio and area under the receiver operating characteristic (ROC) curve by the original and modified STOP-Bang criteria of obstructive sleep apnea by apnea-hyponea index (AHI).**

| Models | Original | Modified |
|---|---|---|
| AHI ≥ 5 | | |
| Odds ratio (95% CI) | 1.50 (1.05, 2.148) | 1.60 (1.17, 2.18) |
| Area under ROC (95% CI) | 62.14 (51.10, 73.18) | 64.11 (52.37, 75.85) |
| AHI ≥ 15 | | |
| Odds ratio (95% CI) | 1.56 (1.07, 2.28) | 1.71 (1.21, 2.40) |
| Area under ROC (95% CI) | 63.32 (51.96, 74.67) | 66.13 (54.05, 78.21) |
| AHI ≥ 30 | | |
| Odds ratio (95% CI) | 1.62 (1.05, 2.50) | 1.72 (1.16, 2.53) |
| Area under ROC (95% CI) | 65.24 (52.56, 77.92) | 68.29 (55.54, 81.04) |

**Notes.**
CI, confidence interval.

The modified STOP-Bang score was created with the mentioned cut-off points for age, body mass index, and neck circumference in Table 3. Both the original and modified STOP-Bang scores were independently predictive of OSA (Table 4). Compared with the original STOP-Bang questionnaire, the modified STOP-Bang score had a higher odds ratio for OSA than the original STOP-Bang score in all three AHI cut-off points (Table 4). For an AHI ≥ 5 events/hour, the modified STOP-Bang score had an odds ratio of 1.60, while the original STOP-Bang score had an odds ratio of 1.50 (Table 4). The area under the ROC curve for the modified STOP-Bang questionnaire was also higher than the original STOP-Bang questionnaire in all three OSA severities. In the severe OSA group, the modified STOP-Bang score had an area under the ROC curve of 68.29%, while the original STOP-Bang had 65.24% (Figs. 2–4).

Both the original and modified STOP-Bang scores were evaluated for their diagnostic properties for OSA by OSA severity (Table 5). The modified STOP-Bang scores had better sensitivity at the cut-off point of 3 for OSA in all three OSA severities than the original one. A modified STOP-Bang score of 3 had a sensitivity of 93.0% for an AHI of ≥ 5 events/hour, 95.9% for an AHI of ≥ 15 events/hour, and 97.6% for an AHI of ≥ 30 events/hour, while an original STOP-Bang score of 3 had sensitivities for these three groups of 50.0%, 53.6%, and 56.1%, respectively. A summary of other diagnostic properties of the original and modified STOP-Bang cut-off point of 3 is shown in Table 6.

## DISCUSSION

This study found that the STOP-Bang criteria for OSA screening in patients with a body mass index of less than 35 kg/m$^2$ had different parameters than the original study (*Chung et al., 2008*). The modified cut-off points from this dataset for age, neck circumference, and body mass index were 40 years, 35 cm, and 23 kg/m$^2$, respectively.

This study found that only the age and hypertension items in the STOP-Bang questionnaire were significantly different between those with and without OSA (Table 1). In contrast, the previous study found that blood pressure, body mass index, age, neck circumference, and sex were significantly different between both groups (*Chung et*

Sinsopa et al. (2025), *PeerJ*, DOI 10.7717/peerj.20310

**Table 5  Diagnostic parameters of the original and modified STOP-Bang cut point for obstructive sleep apnea (OSA) by severity of OSA.**

| Score cut point | Original STOP-Bang | | | | | | Modified STOP-Bang | | | | | |
|---|---|---|---|---|---|---|---|---|---|---|---|---|
| | n (%) | Sense | Spec | PPV | NPV | Youden | n (%) | Sense | Spec | PPV | NPV | Youden |
| AHI ≥ 5, n = 188 | | | | | | | | | | | | |
| 1 | 7 (3.72) | 97.5 | 10.0 | 85.1 | 42.9 | 0.075 | 2 (1.06) | 100 | 6.7 | 84.9 | 100 | 0.067 |
| 2 | 23 (12.23) | 86.7 | 30.0 | 86.7 | 30.0 | 0.167 | 3 (1.60) | 99.4 | 13.3 | 85.8 | 80.0 | 0.127 |
| 3 | 69 (36.70) | 50.0 | 66.7 | 88.8 | 20.2 | 0.167 | 14 (7.45) | 93.0 | 26.7 | 87.0 | 42.1 | 0.197 |
| 4 | 50 (26.60) | 22.2 | 86.7 | 89.7 | 17.4 | 0.088 | 43 (22.87) | 70.9 | 53.3 | 88.9 | 25.8 | 0.242 |
| 5 | 27 (14.36) | 7.0 | 96.7 | 91.7 | 16.5 | 0.036 | 53 (28.19) | 40.5 | 70.0 | 87.7 | 18.3 | 0.105 |
| 6 | 12 (6.38) | NA | NA | NA | NA | NA | 46 (24.47) | 15.8 | 93.3 | 92.6 | 17.4 | 0.092 |
| 7 | 0 | NA | NA | NA | NA | NA | 26 (13.83) | 0 | 96.7 | 0 | 15.5 | 0.000 |
| 8 | 0 | NA | NA | NA | NA | NA | 1 (0.53) | NA | NA | NA | NA | NA |
| AHI ≥ 15, n = 127 | | | | | | | | | | | | |
| 1 | 4 (3.15) | 99.0 | 10.0 | 78.0 | 75.0 | 0.090 | 2 (1.57) | 100 | 6.7 | 77.6 | 100 | 0.067 |
| 2 | 19 (14.96) | 85.6 | 30.0 | 79.8 | 39.1 | 0.156 | 2 (1.57) | 100 | 13.3 | 78.9 | 100 | 0.133 |
| 3 | 42 (33.07) | 53.6 | 66.7 | 83.9 | 30.8 | 0.203 | 8 (6.30) | 95.9 | 26.7 | 80.9 | 66.7 | 0.225 |
| 4 | 35 (27.56) | 23.7 | 86.7 | 85.2 | 26.0 | 0.104 | 28 (22.05) | 75.3 | 53.3 | 83.9 | 40.0 | 0.286 |
| 5 | 19 (14.96) | 7.2 | 96.7 | 87.5 | 24.4 | 0.039 | 37 (29.13) | 42.3 | 70.0 | 82.0 | 27.3 | 0.123 |
| 6 | 8 (6.30) | NA | NA | NA | NA | NA | 32 (25.20) | 16.5 | 93.3 | 88.9 | 25.7 | 0.098 |
| 7 | 0 | NA | NA | NA | NA | NA | 17 (13.39) | 0 | 96.7 | 0 | 23.0 | 0.000 |
| 8 | 0 | NA | NA | NA | NA | NA | 1 (0.79) | NA | NA | NA | NA | NA |
| AHI ≥ 30, n = 71 | | | | | | | | | | | | |
| 1 | 4 (5.64) | 97.6 | 10.0 | 59.7 | 75.0 | 0.076 | 2 (2.82) | 100 | 6.7 | 59.4 | 100 | 0.067 |
| 2 | 9 (12.68) | 90.2 | 30.0 | 63.8 | 69.2 | 0.202 | 2 (2.82) | 100 | 13.3 | 61.2 | 100 | 0.133 |
| 3 | 25 (35.21) | 56.1 | 66.7 | 69.7 | 52.6 | 0.228 | 5 (7.04) | 97.6 | 26.7 | 64.5 | 88.9 | 0.242 |
| 4 | 19 (26.76) | 24.4 | 86.7 | 71.4 | 45.6 | 0.111 | 16 (22.54) | 78.0 | 53.3 | 69.6 | 64.0 | 0.314 |
| 5 | 10 (14.08) | 7.3 | 96.7 | 75.0 | 43.3 | 0.040 | 18 (25.35) | 46.3 | 70.0 | 67.9 | 48.8 | 0.163 |
| 6 | 4 (5.63) | NA | NA | NA | NA | NA | 18 (25.35) | 19.5 | 93.3 | 80.0 | 45.9 | 0.128 |
| 7 | 0 | NA | NA | NA | NA | NA | 9 (12.68) | 0 | 96.7 | 0 | 41.4 | 0.000 |
| 8 | 0 | NA | NA | NA | NA | NA | 1 (1.41) | NA | NA | NA | NA | NA |

**Notes.**

Sense, sensitivity; Spec, specificity; PPV, positive predictive value; NPV, negative predictive value; NA, not available; AHI, apnea-hypopnea index.

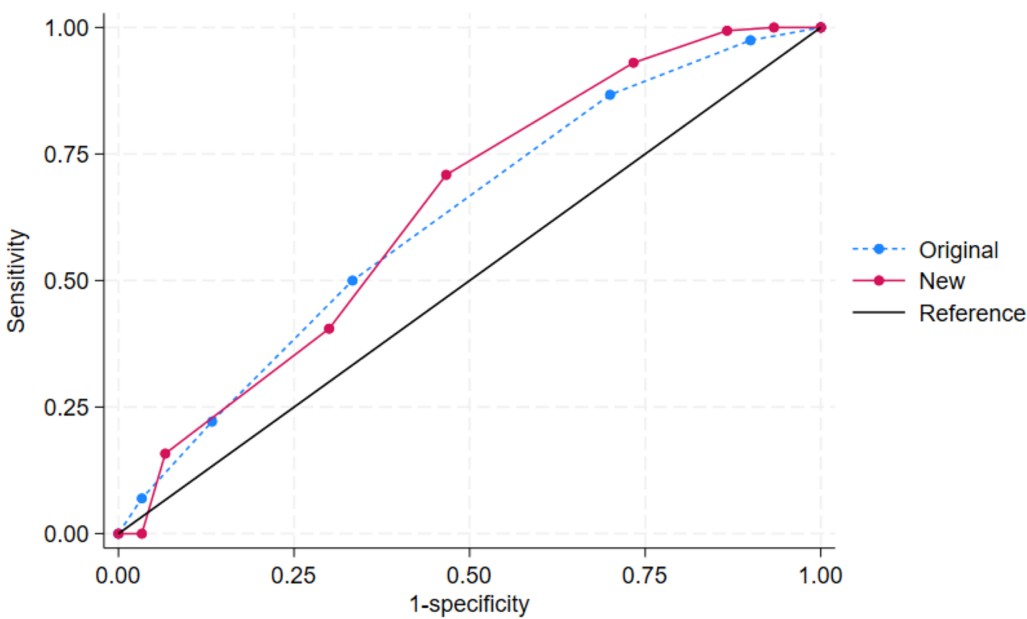

**Figure 2** **The receiver operating characteristic (ROC) curves by the original and new (modified) STOP-Bang criteria of obstructive sleep apnea (OSA): AHI ≥ 5 events/hour.** Note. The area under ROC curve (95% confidence interval) for the original and new (modified) STOP-Bang was 63.32 (51.96, 74.67) and 66.13 (54.05, 78.21), respectively.

al., 2008). After adjusting for factors in the STOP-Bang questionnaire, only age was independently associated with OSA, with an increasing risk of 4% per year (Table 2). Note that the diagnostic criterion for OSA in the previous study was the AHI > 5 events/hour, while this study used the AHI ≥ 5 events/hour. Despite different diagnostic criteria and study populations, this study still found a high prevalence of OSA at 84.04%, which was higher than the previous study's 68.92% (Chung et al., 2008). These findings may be explained by different study populations. The study population in the previous study was preoperative patients, while this study enrolled patients who were suspected of having OSA. This proportion of 84.04% is comparable to previous studies conducted in patients with morning dry mouth (83.1%) or 82.7% in adult patients referred for OSA evaluation at a tertiary care pulmonary clinic (Ma et al., 2023; Öztürk et al., 2019).

The cut-off point for age in the STOP-Bang questionnaire in this study was lower than in the previous study (40 vs. 50 years). The previous study had mean ages for the control and OSA groups of 49 and 58 years, respectively, while the median age of the control group was slightly higher in this study at 51 years, and the median age of the OSA group was similar at 58 years (Table 1). Therefore, this different cut-off point for age between these two studies may be due to the different ages of the study populations. However, it may also be due to body mass index. A previous review found that the prevalence of obesity decreased with age, resulting in a lower cut-off point for age in the questionnaire. Individuals aged 75 years or older had a prevalence of obesity of 27.8%, while those between the ages of 65 and 74 had a higher prevalence of obesity at 40.8% (Zhang et al., 2019).

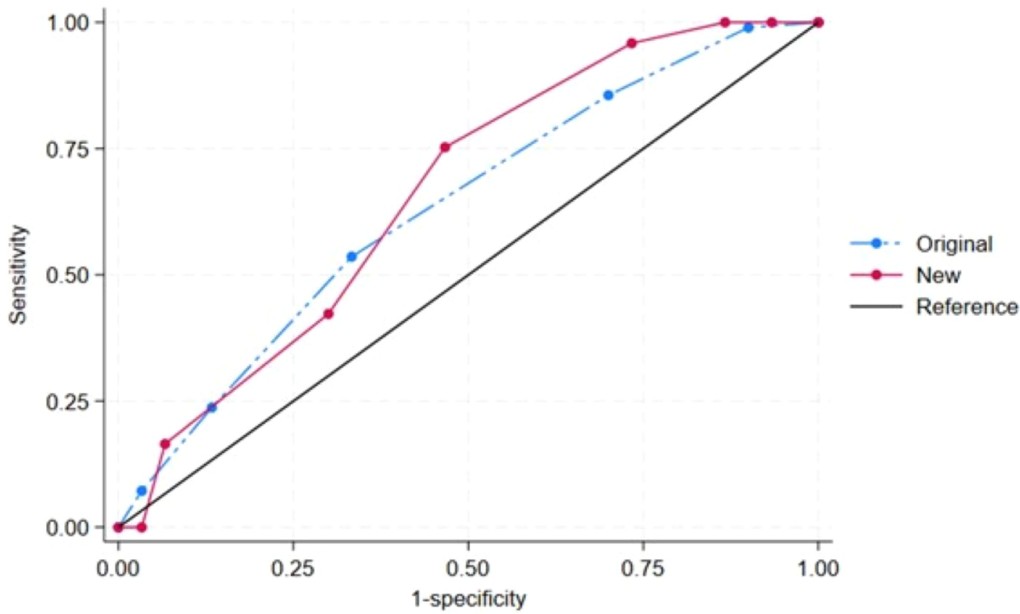

**Figure 3  The receiver operating characteristic (ROC) curves by the original and new (modified) STOP-Bang criteria of obstructive sleep apnea (OSA): AHI ≥ 15 events/hour.** Note. The area under ROC curve (95% confidence interval) for the original and new (modified) STOP-Bang was 63.32 (51.96, 74.67) and 66.13 (54.05, 78.21), respectively.

Several studies showed that neck circumference and body mass index may need to be modified across populations and settings. A study from India found that a neck circumference of 35 cm may be associated with a high sleep score or poor sleep quality (*Sahay et al., 2024*). Patients with OSA and resistant hypertension had a significantly larger neck circumference than those without resistant hypertension: 37.32 *vs.* 35.81 cm; *p*-value <0.001 (*Lin et al., 2024*). Another study found that patients with atrial fibrillation and OSA had a higher body mass index (33.6 *vs.* 31.3 kg/m$^2$; $p = 0.020$) and a larger neck circumference (42 *vs.* 41 cm; $p < 0.001$) than those with atrial fibrillation but without OSA (*Baymukanov et al., 2024*). Even though neck circumference and body mass index might differ between sexes, we used the overall cut-off point for both parameters, as in the original study (*Chung et al., 2012*). The lower cut-off points for neck circumference and body mass index of 35 cm and 23 kg/m$^2$, respectively, in this study may be due to a study population with a body mass index of less than 35 kg/m$^2$.

Another important finding of this study is that the modified STOP-Bang score was created with higher sensitivity to predict OSA in individuals with a body mass index of less than 35 kg/m$^2$, regardless of OSA severity, as shown in Table 5 (*Chung et al., 2008*). The sensitivity of the modified STOP-Bang score was over 90%, ranging from 93.0% to 97.6%, while the original STOP-Bang score had a sensitivity ranging from 50.0% to 56.1% for mild to severe OSA using a cut-off point of 3. These data may imply that parameters, particularly numerical parameters, in the original STOP-Bang questionnaire may need modification according to body mass index. A previous study confirmed this hypothesis,

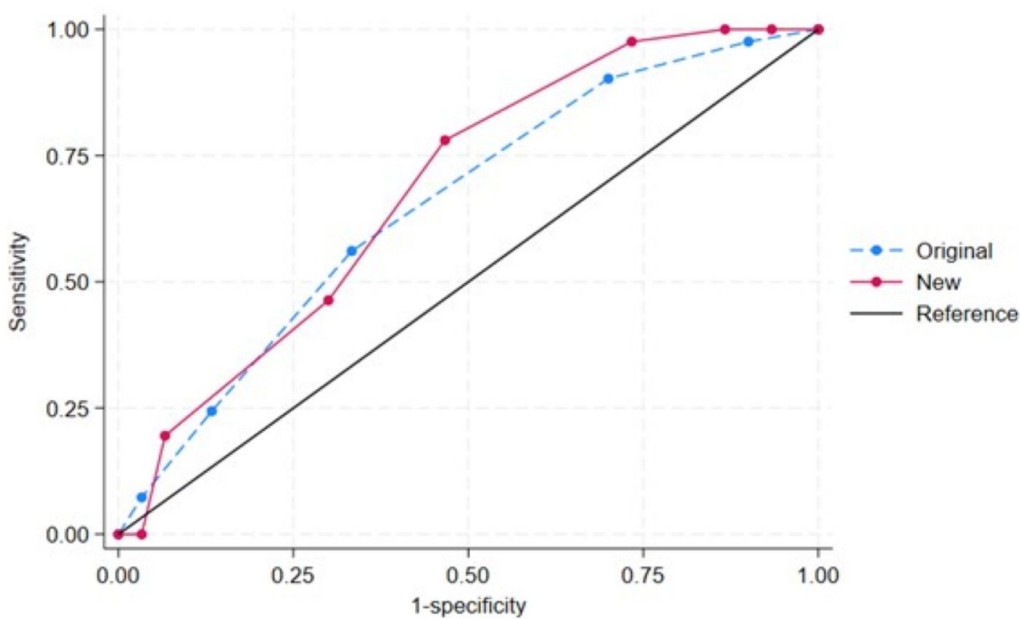

**Figure 4** **The receiver operating characteristic (ROC) curves by the original and new (modified) STOP-Bang criteria of obstructive sleep apnea (OSA): AHI ≥ 30 events/hour.** Note. The area under ROC curve (95% confidence interval) for the original and new (modified) STOP-Bang was 65.24 (52.56, 77.92) and 68.29 (55.54, 81.04), respectively.

as body mass index was significantly associated with age and neck circumference, with Spearman correlations of 0.28 ($p = 0.001$) and 0.59 ($p < 0.001$), respectively (*Dos Santos et al., 2022*). Additionally, the results of this study may apply to those who are suspected of having OSA in a real-world clinical practice setting, not just the preoperative setting (*Chung et al., 2008*). The modified STOP-Bang questionnaire may be more appropriate for those with a body mass index of less than 35 kg/m$^2$. This may result in a more sensitive tool for selecting suspected patients for polysomnography in primary care or resource-limited settings. Early detection of OSA may result in lower cardiovascular risks in the future.

Some studies have examined a modified STOP-Bang questionnaire (*Waseem et al., 2021*; *Loh & Toh, 2019*; *Xia et al., 2018*). These studies showed different cut-off points from the original STOP-Bang questionnaire, such as a body mass index of 27.5 or 28 kg/m$^2$ or a neck circumference of 35 cm. Unlike this study, those studies included patients with a body mass index of over 35 kg/m$^2$. Another study showed that the waist-to-hip ratio may be a better parameter than neck circumference, particularly in female patients (*Lim et al., 2014*). However, the waist-to-hip ratio was not used to calculate the sensitivity for OSA diagnosis in the STOP-Bang questionnaire. Note that there is very limited STOP-Bang questionnaire for individuals with low body mass index setting. In other words, this study may be the very first study on this aspect.

There were 188 patients in this study which was larger than the estimated sample size of 154 patients. The *post-hoc* power of this study population compared with the previous study had the power of 100% (*Chung et al., 2008*). There are some limitations to this study.

**Table 6** Summary of diagnostic properties of the original and modified STOP-Bang cut point of 3 for obstructive sleep apnea (OSA) by severity of OSA.

| Factors | Original STOP-Bang | Modified STOP-Bang |
| --- | --- | --- |
| All OSA (AHI ≥ 5) | | |
| AUC | 62.14 | 64.11 |
| Sensitivity | 50.0 | 93.0 |
| Specificity | 66.7 | 26.7 |
| PPV | 88.8 | 87.0 |
| NPV | 20.2 | 42.1 |
| Moderate OSA (AHI ≥ 15) | | |
| AUC | 63.32 | 66.13 |
| Sensitivity | 53.6 | 95.9 |
| Specificity | 66.7 | 26.7 |
| PPV | 83.9 | 80.9 |
| NPV | 30.8 | 66.7 |
| Severe OSA (AHI ≥ 30) | | |
| AUC | 65.24 | 68.29 |
| Sensitivity | 56.1 | 97.6 |
| Specificity | 66.7 | 26.7 |
| PPV | 69.7 | 64.5 |
| NPV | 52.6 | 88.9 |

Notes.

AHI, apnea-hypopnea index; AUC, area under the receiver operating characteristic (ROC) curve; PPV, positive predictive value; NPV, negative predictive value.

First, our study had an upper limit for body mass index of 35 kg/m$^2$ and enrolled suspected OSA patients at the outpatient department of a single center with small sample size, which may result in selection bias. The high OSA prevalence may inflate sensitivity and affect generalizability. Therefore, the results of this study may be applicable to the outpatient setting at university hospitals. Further studies may be required in other settings, such as community-based populations, to confirm generalizability, along with a validation study using a multi-center, larger sample size. Second, other risk factors or comorbidities, such as smoking, were not included in the analysis as they were not included in the original study (*Chung et al., 2008*). The neck circumference in this modified STOP-Bang was uniform for both sexes, as previously reported in the earlier version of the STOP-Bang questionnaire (*Chung et al., 2012*). This may affect its accuracy. Only hypertension was included in the study, while other comorbidities such as COPD or neuromuscular diseases were not excluded. This may affect the generalizability of the results. The model was computed without establishing a causal relationship or using directed acyclic graphs (*Menezes-Júnior et al., 2024*). Finally, no intervention, such as a continuous positive airway pressure machine, was implemented (*Kaewkes, Sawanyawisuth & Sawunyavisuth, 2020*; *Sawunyavisuth, Ngamjarus & Sawanyawisuth, 2023*). Prospective studies of this modified STOP-Bang questionnaire on CPAP adherence or long-term clinical endpoints may be warranted.

## CONCLUSIONS

The STOP-Bang questionnaire may need to be modified for individuals with a body mass index of less than 35 kg/m$^2$ who are suspected of OSA. The cut-off points for age, neck circumference, and body mass index were 40 years, 35 cm, and 23 kg/m$^2$, respectively. In patients with BMI less than 35 kg/m$^2$, the modified STOP-Bang substantially improves sensitivity and therefore may be a more reliable screening tool in this subgroup.

### Funding
The authors received no funding for this work.

### Competing Interests
The authors declare there are no competing interests.

### Author Contributions
- Napassorn Sinsopa conceived and designed the experiments, performed the experiments, prepared figures and/or tables, and approved the final draft.
- Viriya Tripakornkusol conceived and designed the experiments, performed the experiments, prepared figures and/or tables, and approved the final draft.
- Sittichai Khamsai conceived and designed the experiments, performed the experiments, analyzed the data, prepared figures and/or tables, authored or reviewed drafts of the article, and approved the final draft.
- Kittisak Sawanyawisuth conceived and designed the experiments, performed the experiments, analyzed the data, prepared figures and/or tables, authored or reviewed drafts of the article, and approved the final draft.

### Human Ethics
The following information was supplied relating to ethical approvals (*i.e.*, approving body and any reference numbers):

The study protocol was approved by the ethics committee in human research, Khon Kaen University, Thailand (HE641504).

### Data Availability
The raw data is available in the Supplementary File.

The code definitions for the categorical data are described in the raw data file.

### Supplemental Information
Supplemental information for this article can be found online at http://dx.doi.org/10.7717/peerj.20310#supplemental-information.

![PeerJ]

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
