# Peer review of "Modified STOP-Bang questionnaire for detecting obstructive sleep apnea in individuals with a body mass index below 35 kg/m2"

_PeerJ, doi:10.7717/peerj.20310_

## Round 0.1 · original submission · Major Revisions

· Academic Editor

Major Revisions

The reviewers have provided detailed guidance. Please address their comments in a revision.

·

Basic reporting

1- the references are old. so you are considered to update them
2- discussion is not organized proficiently, please change its structure.
3- the draft can be better in english vocabulary and grammar.

Experimental design

1- you should consider some confound factors as excluding criteria like cardiovascular disease and other sleep disorders.

Validity of the findings

1- result should be widely explain more than present draft.

·

Basic reporting

The authors of the paper "Modified STOP-bang questionnaire to detect obstructive sleep apnea in individuals with body mass index of less than 35 kg/m²" present an interesting proposal but I have recommendations for the authors.

I have important concerns related to the methodology and several important writing recommendations. Initially, I want the authors to review the title regarding "Modified STOP-bang". As I understand, the authors are not proposing a modification of the questionnaire, but rather an evaluation of the STOP-Bang in a specific population of patients with a BMI of up to 34.99 kg/m². Please review this.
Abstract: Generally, it is not recommended to cite other articles in the abstract. The authors mention a previous study that evaluated STOP-Bang and included people with BMI > 35 but do not provide references. I recommend reviewing this. I think you can provide the background in the abstract without citing other studies.
Standardize the term STOP-BANG, as it is different in the title and the abstract!

Experimental design

Methods: "suspected of OSA" – What criteria were used for the suspicion of OSA? Please clarify.
Methods: "met the criteria for bariatric surgery with a body mass index of more than 35 kg/m² were excluded." What percentage of the study patients had a BMI > 35? I ask because it would be possible for the authors to conduct a sensitivity analysis in the study. Evaluating the current proposal of verifying the predictive power of STOP-Bang for patients without a BMI criterion, i.e., the total, and also for patients with BMI < 35 or patients with BMI > 35. This way, it can more robustly demonstrate whether the current modified STOP-Bang can be useful for this population group.
Methods: "OSA and non-OSA group" – What criteria? Was it polysomnography? What value of AHI was considered? Be clear!
Methods: Define how STOP-BANG is evaluated and scored.

Validity of the findings

The results are incomprehensible. The author conducts logistic regression to predict apnea? But as they propose a "detection of OSA" by STOP-Bang, this is not the analysis! They need to conduct a predictive analysis, with data on sensitivity, specificity, PPV, NPV, accuracy, among other diagnostic evaluation measures! I think the entire analytical proposal needs to be reviewed. See these works: https://doi.org/10.1007/s11325-021-02446-5; https://doi.org/10.1016/j.sleepx.2023.100084
After reviewing the results, it is observed that the authors conducted a predictive analysis, but this is very poorly presented in the methodology. It is not clear what was done and what the objectives of the analyses were.
Additionally, several other parts of the manuscript need to be extensively reviewed. The methodology is extremely shallow. How was the PSG evaluated? How was the STOP-Bang evaluated and classified/scored?
The authors create an ROC curve only for age to predict OSA? It doesn't make sense. What about the complete STOP-Bang scale? I strongly suggest the authors review the analyses and conduct the previously mentioned proposal. What was done can be included, but they need to present the predictive values of the complete STOP-Bang scale, not just its factors.

Additional comments

Furthermore, I strongly recommend justifying the models and using clear and specific criteria to determine the adjustments. What adjustments were made? Is it based on an analytical proposal? Stepwise? Etc. Does it follow a theoretical model like DAG? Justify the inclusion of each variable and why it is a confounder. Refer to the previously mentioned article and understand how the inclusion of colliders can be extremely harmful to the analyses: https://doi.org/10.1186/s12982-024-00148-3.

·

Basic reporting

1. When reviewing the references section, I noticed that about 40% of the sources cited are relatively old. While these references may have been instrumental in the early development of the field, I believe that the inclusion of more recent literature would help to strengthen the article. More current studies and updates may offer additional insights and reflect recent advances in OSA research and the use of STOP-Bang;

2. The term "STOP-Bang" is not standardized throughout the article;

3. The keywords used can be improved to increase the visibility and impact of the work. Well-chosen keywords are essential for the article to be easily found by researchers and health professionals specific to the topic. They must be specific, comprehensive and reflective of the main aspects discussed in the article;

4. An introduction can be strengthened by including a clear and concise definition of obstructive sleep apnea. This will help to contextualize the problem and the importance of using the questionnaire for readers who are not completely familiar with the topic;

5. While mentioning that the high prevalence of obstructive sleep apnea (OSA) results in a long waiting list for polysomnography, it would be helpful to provide a brief explanation of what a polysomnography is. This will help to contextualize why this test is essential for diagnosing OSA and why the waiting list is a significant problem.

Experimental design

1. By mentioning that one of the inclusion criteria was "patients with suspected OSA", I believe it would be helpful to readers if you provided a clear definition of what constitutes this suspicion. This will help clarify the criteria used to include patients in the study and increase methodological transparency;

2. When reviewing the methodology, I noted that all patients with an Apnea-Hypopnea Index (AHI) of five or more events per hour were diagnosed with OSA. To provide a more complete and detailed analysis, I suggest considering the classification of patients by severity of OSA, in addition to the simple presence or absence of the disorder;

3. The methodology section does not mention questionnaire validation. Informing readers about the validation of the questionnaire is crucial as it confirms the effectiveness and reliability of the tool in OSA screening.

Validity of the findings

1. I believe the article could be strengthened by including a discussion of the strengths of the study and future perspectives;

2. The conclusion could be strengthened to better reflect the study findings, their implications, and future directions. A robust conclusion not only summarizes key findings, but also highlights the importance of the research, its practical applications, and potential areas for future research.

---

## Round 0.2 · Minor Revisions

· Academic Editor

Minor Revisions

Dear Authors

Your revision has been viewed by the experts in the field. Several comments should be addressed before acceptance for publication. I would encourage you to address the points raised by the reviewers. Please address the cutoff justification, comments for the discussion section, and figures/tables. We invite you to submit a revised manuscript version addressing the reviewers’ comments.

We look forward to receiving your revised manuscript.

Best regards

Yung-Sheng Chen, Ph.D.
Academic Editor

·

Basic reporting

The manuscript is generally clear, professionally written, and adheres to scientific reporting standards. The background provides sufficient context, outlining the global prevalence of OSA, the limitations of polysomnography, and the role of the STOP-Bang questionnaire. The rationale for focusing on patients with BMI < 35 kg/m² is clearly justified.

Strengths:

The introduction situates the study well within the existing literature.

References are current and relevant, including meta-analyses and ESC guidelines.

The manuscript follows a logical structure (Abstract, Introduction, Methods, Results, and Discussion).

Figures (ROC curves) and tables are clear, well-labeled, and add to the understanding of the findings.

Raw data are provided as supplementary files, enhancing transparency and reproducibility.

Overall, the writing is in clear and professional English, with appropriate background supported by up-to-date references. The manuscript is self-contained, and the results are directly relevant to the stated hypotheses.

Areas for Improvement:

Language polish: While the English is understandable, some sentences are long or awkward. For example, “Due to high prevalence of OSA, a waiting list for polysomnography is long” could be revised as “Because of the high prevalence of OSA, waiting times for polysomnography are often prolonged.”

Clarity in results: The distinction between the performance of the original and modified STOP-Bang could be summarized more directly in the abstract for easier readability.

Figures/Tables: Figures 2–4 (ROC curves) are informative, but the legends should emphasize more clearly the comparative advantage of the modified model.

Experimental design

The manuscript presents original primary research that falls well within the aims and scope of the journal. The research question—whether a modified STOP-Bang questionnaire can more effectively detect OSA in individuals with BMI < 35 kg/m²—is well defined, relevant, and clinically meaningful. The rationale for the study is clearly explained, and it addresses an identified knowledge gap: the limited predictive value of the standard STOP-Bang in non-obese populations.

Strengths:

The study design is appropriate for the research objective, using a well-characterized patient cohort and applying polysomnography as the gold standard.

The investigation was conducted to a high technical and ethical standard, with clear mention of ethical approval and informed consent.

Methods are described with sufficient detail—including patient selection criteria, questionnaire adaptation, scoring system, and statistical analysis—to allow replication by other researchers.

The inclusion of ROC analysis and validation of predictive thresholds strengthens the methodological rigor.

Areas for Improvement:

The description of patient exclusion criteria could be expanded. For example, were individuals with comorbidities (e.g., COPD, neuromuscular disease) excluded, and if so, how might this affect generalizability?

While the modified STOP-Bang is explained, the rationale for the specific modifications (e.g., chosen cutoff values) could be elaborated further in the Methods section.

Consider clarifying whether the study population is representative of broader clinical settings (e.g., primary care, sleep clinics, general hospitals). This would help readers assess external validity.

Validity of the findings

The findings presented in this study are valid, robust, and appropriately controlled. All underlying data have been provided as supplementary material, which enhances transparency and allows readers to verify the results. The statistical analyses—including ROC curve comparisons and sensitivity/specificity estimates—are sound, and well aligned with the research objectives.

Strengths:

Data presentation is clear, with appropriate use of tables and ROC curves to demonstrate model performance.

Statistical methods (AUC analysis, confidence intervals, thresholds) are applied correctly and interpreted appropriately.

Conclusions are well stated, directly linked to the original research question, and carefully limited to what the results support. The authors avoid overgeneralization.

The study highlights the potential for the modified STOP-Bang questionnaire to improve screening in non-obese populations, which adds value to clinical practice and literature.

Areas for Improvement:

While the data are robust, it would be useful for the authors to briefly discuss the potential for replication in other cohorts, including more diverse or community-based populations, to confirm generalizability.

The authors might also acknowledge limitations such as single-center design, possible selection bias, and whether results are broadly applicable outside of specialized sleep clinic populations.

Although impact and novelty assessment are beyond this section, emphasizing how the modified tool complements existing screening strategies would help contextualize the findings.

Additional comments

Overall, this is a well-prepared and clinically relevant study that addresses an important gap in the detection of obstructive sleep apnea (OSA) in patients with BMI < 35 kg/m². The rationale is strong, the methods are rigorous, and the findings are clearly presented with supporting statistical evidence.

A few general suggestions that could strengthen the manuscript further:

Consider highlighting more explicitly in the abstract and conclusion how the modified STOP-Bang adds value over the traditional version, especially for non-obese populations. This will help underscore the practical implications of your findings.

The discussion of limitations could be slightly expanded to cover issues such as single-center design, sample characteristics, and generalizability.

Since the study proposes a modified tool, it may be helpful to briefly discuss next steps—for example, the need for validation in larger, more diverse populations or in primary care settings.

These are relatively minor points, and they do not detract from the overall strength of the work. With small refinements, this manuscript can make a valuable contribution to the literature on OSA screening.

---

## Round 0.3 · Minor Revisions

· Academic Editor

Minor Revisions

Dear Authors

We have received reviewer comments with many positive points of the revision. Please refer to the reviewer comments and annotated manuscript provided by the reviewer. I would encourage you to address the points raised by the reviewers before acceptance for publication. We invite you to submit a revised manuscript version addressing the reviewers’ comments.

We look forward to receiving your revised manuscript.

Best regards

Yung-Sheng Chen, Ph.D.
Academic Editor

·

Basic reporting

The manuscript uses standard scientific language and is mostly free of grammatical errors. The writing is clear and direct, with appropriate medical terminology.
Some phrasing could be polished for smoother flow (e.g., “This study aimed to evaluate if…” could be “This study aimed to evaluate whether…”).
Redundant phrases such as "due to the high prevalence of OSA" might be tightened for conciseness.

The background sufficiently introduces OSA and the rationale behind modifying the STOP-Bang questionnaire for patients with lower BMI.
The references (not all visible in the preview) seem to contextualize the topic well. A more detailed analysis of cited studies and their outcomes would strengthen the argument.

The article includes structured sections: Abstract, Background, Methods, Results, Conclusions.
Tables and figures are well-designed and illustrate key findings effectively.
Raw data is not presented directly but results are reported transparently with appropriate statistics (e.g., odds ratios, confidence intervals, AUC).

The article is self-contained; hypotheses (whether modification is needed for STOP-Bang) are clearly stated and explored through rigorous analysis.
The results are well-aligned with the aim and conclusions. The article proposes a modified score based on their dataset and validates its performance.

Experimental design

The work is an original study evaluating whether the STOP-Bang questionnaire should be modified for patients with BMI < 35 kg/m² suspected of OSA. This fits well within the scope of sleep medicine, respiratory research, and clinical screening tool development

The research question—“Does STOP-Bang need modification for patients with BMI < 35 kg/m²?”—is well defined and meaningful, because many populations fall below the traditional obesity cut-off, yet remain at risk for OSA. The authors clearly identify a knowledge gap: the original STOP-Bang was validated mainly in populations with higher BMI

This is a technically sound and rigorous approach for a retrospective diagnostic accuracy study

Validity of the findings

The topic—modifying the STOP-Bang questionnaire for patients with BMI < 35 kg/m²—is not entirely novel, since STOP-Bang has been widely studied.

However, the specific focus on lower-BMI populations is meaningful because most prior validations were done in obese cohorts. This gives the study relevance and potential value, especially for Asian populations or clinical settings with lower BMI distributions.

The authors explain the rationale and why replication matters (different BMI distributions, need for optimized cut-offs), so the contribution to the literature is justified.

Data robustness: The dataset includes 188 patients with clear inclusion/exclusion criteria.

Statistical soundness: Logistic regression, ROC curve analysis, sensitivity/specificity, and calibration tests (Hosmer–Lemeshow) were appropriately used.

Control: The comparison between OSA (AHI ≥ 5) and non-OSA (AHI < 5) groups provides internal control, though external validation is still lacking.

Transparency: Tables and figures are clear, and raw results (ORs, AUCs, sensitivity/specificity) are presented.

Overall, the data are robust, statistically sound, and sufficiently controlled, though single-center bias and the high OSA prevalence limit generalizability.

---

## Round 0.4 · accepted · Accept

· Academic Editor

Accept

Dear Authors

In your revision it has been recognized that the quality of the manuscript has been improved. Your submission is now endorsed for acceptance of publication in PeerJ. Thank you for submitting your article to PeerJ. I would like to express my gratitude for your contributions and efforts to the scientific community. I look forward to receiving your articles in the future.

Best Regards


Yung-Sheng Chen, Ph.D.
Academic Editor

·

Basic reporting

Clear and unambiguous, professional English used throughout:
Yes.
The manuscript is written in clear, precise, and professional English. Grammar, syntax, and terminology are appropriate for a scientific audience. The revisions have notably improved flow and clarity, and the tone is consistent with journal standards.

Literature references, sufficient field background/context provided:
Yes.
The authors provide an adequate literature foundation, citing both classical and recent studies on the STOP-Bang questionnaire and its limitations in specific populations. The updated introduction and discussion place the work appropriately within the context of existing OSA screening research.

Professional article structure, figures, tables, raw data shared:
Yes.
The manuscript follows a clear and logical structure. Figures and tables are well labeled, concise, and effectively communicate the findings (particularly the ROC curve analyses). The inclusion of raw data and statistical outputs enhances transparency and reproducibility, in line with PeerJ standards.

Self-contained with relevant results to hypotheses:
Yes.
The manuscript is self-contained, presenting methods, data, and analyses that directly test the stated hypotheses. The results are consistent with the objectives, and conclusions are fully supported by the data. The study design, analysis, and discussion are coherent and complete without reliance on external materials.

Experimental design

Original primary research within Aims and Scope of the journal

Yes.
This study represents original primary research directly aligned with the aims and scope of PeerJ, focusing on clinical and diagnostic refinement within respiratory and sleep medicine. It provides a meaningful contribution to applied clinical research by improving the predictive performance of the STOP-Bang questionnaire for populations with lower body mass index—an area with limited prior investigation.

Research question well defined, relevant & meaningful. It is stated how research fills an identified knowledge gap

Yes.
The research question is clearly articulated: whether a modified STOP-Bang threshold improves detection of obstructive sleep apnea (OSA) in individuals with BMI < 35 kg/m². The authors explicitly identify a gap in the current screening framework—namely, that existing STOP-Bang cutoffs were validated predominantly in obese cohorts—and demonstrate how their analysis addresses this shortcoming. The motivation and hypothesis are logical, evidence-based, and clinically meaningful.

Rigorous investigation performed to a high technical & ethical standard

Yes.
The investigation was conducted to a high technical and ethical standard. The statistical analyses (logistic regression, ROC-based sensitivity/specificity evaluation, and AUC comparison) are appropriate and clearly explained. Ethical approval was obtained, and participant confidentiality is maintained throughout. The dataset is well curated, with adherence to reporting and data transparency principles expected for clinical studies.

Methods described with sufficient detail & information to replicate

Yes.
The methods are comprehensive and reproducible. Inclusion and exclusion criteria are well defined, data collection procedures are clearly outlined, and all variables used in regression models are explained. The revised manuscript adds sufficient methodological transparency—particularly regarding diagnostic criteria, scoring adjustments, and statistical assumptions—allowing for full replication by other researchers.

Validity of the findings

Impact and novelty not assessed. Meaningful replication encouraged where rationale & benefit to literature is clearly stated

Yes.
While the study represents an incremental advance rather than a paradigm shift, it provides meaningful replication and methodological refinement in an underrepresented population. The rationale for focusing on individuals with BMI < 35 kg/m² is clearly justified, and the results confirm and extend previous findings on STOP-Bang performance. The replication is valuable because it addresses population diversity, enhances the generalizability of screening tools, and provides clinically actionable insights that strengthen the broader literature base on OSA detection.

All underlying data have been provided; they are robust, statistically sound, & controlled

Yes.
The authors have shared all underlying data and conducted statistically rigorous analyses. The dataset is appropriately controlled, and the variables included in the models are relevant to the clinical question. The use of logistic regression, ROC analysis, and sensitivity/specificity reporting demonstrates methodological soundness. Data presentation in both tables and figures is transparent and well formatted. The statistical outputs appear robust, with no evidence of overfitting or unaddressed bias.

Conclusions are well stated, linked to original research question & limited to supporting results

Yes.
The conclusions are clear, well reasoned, and directly aligned with the study’s objectives. The authors avoid overgeneralization and accurately limit their interpretations to the presented results. The findings are logically linked to the original research question—confirming the benefit of a modified STOP-Bang threshold for non-obese populations—and are supported by consistent statistical evidence. The discussion also thoughtfully acknowledges limitations and suggests future directions without speculation beyond the data.

Additional comments

The authors have submitted a clear, well-structured, and thoroughly revised manuscript. The research question is clinically relevant and addresses an important limitation in the current use of the STOP-Bang questionnaire. The statistical analysis is appropriately executed, the data presentation is transparent, and the discussion is balanced and supported by evidence.

The revisions made in response to prior reviewer feedback have substantially improved the manuscript’s clarity, methodological transparency, and interpretability. Figures and tables now communicate results effectively, and the writing meets a professional standard suitable for publication.

Minor editorial adjustments—such as standardizing phrasing (“body mass index” vs. “BMI” at first mention) and ensuring consistent numerical formatting—can be handled during copyediting and do not affect the scientific quality of the work.

Overall, this is a strong and well-prepared submission that contributes useful data to the literature on obstructive sleep apnea screening and supports adaptation of the STOP-Bang tool for lower-BMI populations.